# Arc Characteristics and Welding Process of Laser K-TIG Hybrid Welding

Hongchang Zhang [1,2], Jiang Yu [3], Zixiao Zhang [3], Jianguo Gao [4], Zhaofang Su [4], Zhaorong Sun [5] and Yinan Li [1,*]

[1] School of Mechanical and Automotive Engineering, Qingdao University of Technology, Qingdao 266520, China; hongchangzhang123@163.com
[2] School of Rongcheng, Harbin University of Science and Technology, Weihai 264300, China
[3] State Key Laboratory of Advanced Welding and Joining, Harbin Institute of Technology, Harbin 150001, China; yujianghit@163.com (J.Y.); 15821516518@163.com (Z.Z.)
[4] Shandong Classic Group Co., Ltd., Jining 272000, China; gaojianguo@jdjt.com (J.G.); 972144899@163.com (Z.S.)
[5] Shandong Jinrunde New Material Technology Co., Ltd., Zibo 255000, China; zhrs69@163.com
* Correspondence: liyinan@qut.edu.cn

**Abstract:** The Q235 steel plate butt joint was successfully welded by the laser K-TIG hybrid welding method. The effects of hybrid welding process parameters such as welding current, the distance between heat sources, laser power, laser defocusing amount, and welding speed on the coupled arc profile and welding process stability were studied. The results indicated that the laser deflects the K-TIG arc, and the deflection angle becomes smaller as the arc current increases. After K-TIG generates small holes, if the laser beam acts on the bottom of the keyhole, the welded depth can be further increased; however, the laser power has little effect on the welded depth. The distance between heat sources is the main factor affecting the state of laser-arc coupling. Optical microstructures of welded joints showed that the grains in the arc zone were coarser than those in the laser zone, and there are more columnar crystals in the fusion zone. The microhardness of the weld center is significantly higher than that of the base metal, up to 220 HV. At the same time, the change of tensile strength of the weld under the influence of a single parameter was analyzed, and it was found that tensile properties of the weld first increased and then decreased with the increase of K-TIG arc current $I$, heat source distance $D$, and welding speed $V$, respectively. With the increase of laser power $P$, it first decreased and then increased, and with the increase of laser defocusing amount $\delta f$, it showed a downward trend.

**Keywords:** laser K-TIG hybrid welding; coupling arc profile; process parameters; microstructure analysis; tensile strength



## 1. Introduction

Hybrid welding includes hybrid laser-arc welding, hybrid plasma-MIG/MAG welding, and other technologies [1,2]. Hybrid laser-arc welding is widely used in the field of marine engineering. Laser-arc hybrid welding technology is a high-efficiency welding method that combines two heat sources with completely different physical properties and energy transmission mechanisms, acting on the same processing position, to achieve complementary advantages [3–6]. In recent years, many scholars have studied the mechanism of laser-arc interaction and found that the laser can compress and guide the arc [7]. At the same time, the arc can strengthen the laser, dilute the plasma, and improve the absorption rate of the metal to the laser [8–11]. Laser-arc hybrid welding has the advantages of two heat source welding methods of laser and the conventional arc welding, and the interaction between the two heat sources makes up for the defects of a single heat source application. Compared with single arc welding, laser arc hybrid welding has the characteristics of large penetration, high efficiency, low heat input, and small deformation [12,13]. Due to the stability of the TIG arc, the combination of laser and TIG is a common form of hybrid

welding [14–17]. However, its tungsten electrode has the low current-carrying capacity [18], shallow single-pass penetration, and many composite welding process parameters, which are not conducive to coordinated adjustment and control of weld formation [19]. Therefore, further optimization of process parameters is urgently needed.

Keyhole TIG (K-TIG) welding increases the carrying current and reduces the area of the arc cathode area by adding a water cooling system to the tungsten electrode torch, so as to improve the arc energy density and arc pressure [20–23]. When the welding current is large enough, the arc pressure is greater than the surface tension of the liquid molten pool, and small holes are formed inside the molten pool [24–27]. If the penetrating power of the arc is strong enough, the small hole will completely penetrate the workpiece. Without any filler material, single-pass full penetration welding of 5–13 mm thick plates can be achieved [28]. Compared with TIG welding, KTIG has the advantages of carrying a large welding current, large penetration, and fast welding speed. Theoretically, the combination of laser and K-TIG will have the characteristics of fast welding speed, large penetration, and strong bridging ability. In addition, there are few research reports on laser K-TIG hybrid welding at home and abroad.

Therefore, this paper proposed the laser K-TIG hybrid welding technology and used Q235 carbon steel as the material to carry out the flat-plate butt test. The influence of the hybrid welding process parameters on the coupled arc profile, the microstructure, and the mechanical properties of the welded joints were discussed.

## 2. Experimental System and Procedure

### 2.1. Experimental System

The test base metal is a Q235 steel plate with good weldability, and the specimen dimensions were 200 mm × 100 mm × 5 mm. Using the plate butt test method, the chemical composition and mechanical properties of the base metal are described in Tables 1 and 2, respectively.

**Table 1.** Q235 base metal chemical composition table (wt.%).

| Material Name | Quality Level | C | Mn | Si | S | P |
|---|---|---|---|---|---|---|
| Q235 | B | <0.22 | 0.3–0.65 | <0.35 | <0.050 | <0.045 |

**Table 2.** Q235 Mechanical properties.

| Material Name | Yield Strength/MPa | Tensile Strength/MPa | Elongation % | Impact Energy Akv/J |
|---|---|---|---|---|
| Q235 | 235 | 370–500 | ≥20 | ≥29 |

Figure 1 depicted the schematic diagram of laser K-TIG welding systems, including the YAG2400 W laser power source, K-TIG welding power source WSME-630, self-designed K-TIG welding torch, CCD camera, and welding robot. WSME-630 adopts a unified adjustment method. The arc welding voltage is automatically adjusted as the arc welding current changes. The robot welding platform with a self-designed hybrid welding adjustment device can not only control the opening and closing of the laser and the K-TIG welding torch but also the laser K-TIG hybrid welding process parameters.

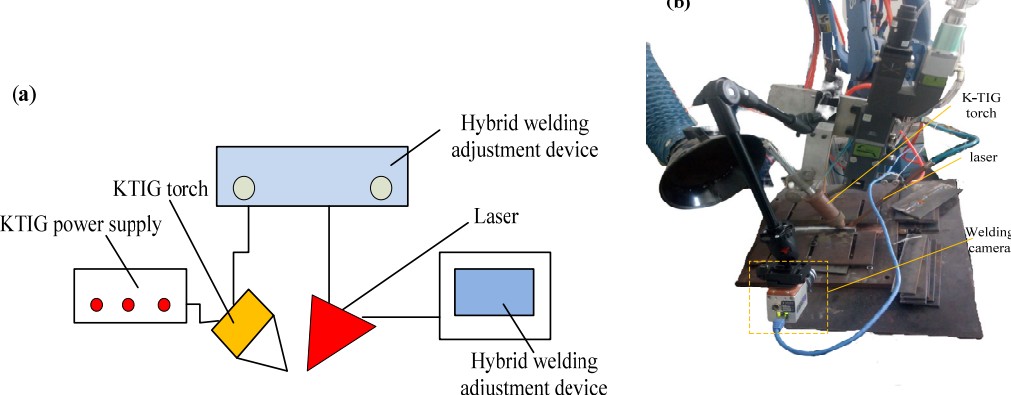

**Figure 1.** Laser K-TIG hybrid welding system: (**a**) schematic diagram, (**b**) structural hybrid diagram.

*2.2. Laser K-TIG Hybrid Welding Process Parameters*

According to the relative position of the arc and the laser, the hybrid method is divided into K-TIG laser hybrid welding with arc-guided laser and laser K-TIG hybrid welding with the laser-guided arc. This system adopts the method of laser in front with the purpose of preheating the workpiece and producing keyhole, and K-TIG arc in the rear during the welding process. K-TIG arc adopts DC positive connection. Argon (80 vol-%) and carbon dioxide (20 vol-%) were used as shielding gas with a flow rate of 12 L/min. The Vickers microhardness was measured via the hardness tester (HV-1000) (Minghu Inc., Jinan, China) with a dwell time of the 20 s and applied a load of 1.96 N. The tensile test was conducted with a microcomputer control electron universal testing machine (WD-P4504) (Chntestm Inc., Jinan, China) at room temperature using a displacement rate of 1 mm/min to evaluate the welded joint properties. The horizontal distance between the laser action point and the tip of the tungsten electrode is defined as the heat source distance $D$. The height between the tungsten electrode and the workpiece surface is the height $H$ of the tungsten electrode. As shown in Figure 2a, the angle between the axis of the welding torch and the horizontal plane is 45°. The angle between the axis of the laser and the horizontal plane is 85°. The dimension of tensile specimens is shown in Figure 2b.

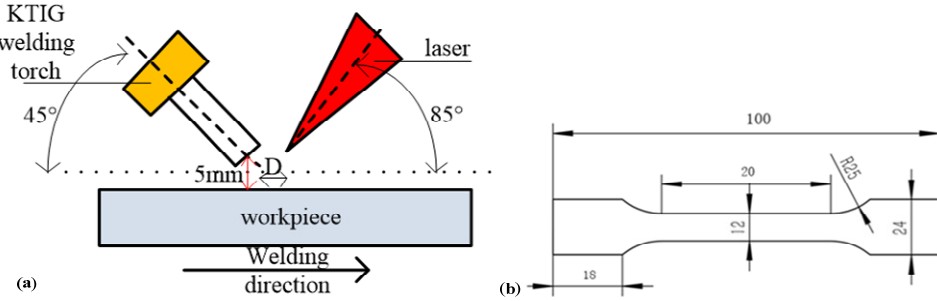

**Figure 2.** Schematic diagram: (**a**) the distribution of hybrid welding parameters. (**b**) tensile samples.

After the sample is fixed and pressed, the coordinated control of the above process parameters alone or with each other can be achieved through the hybrid welding adjustment device. The laser K-TIG hybrid welding process parameters mainly include heat source spacing $D$, welding speed $V$, K-TIG arc current $I$, laser power $P$, and defocus amount $\delta f$, and orthogonal experiments are used to analyze the influence of the above parameters on the coupled arc profile and weld formation. The detailed parameters of hybrid welding are shown in Table 3.

**Table 3.** Detailed hybrid welding parameters.

| Parameter Set | K-TIG Current/A | Laser Power/W | Defocus Amount/mm | Heat Source Spacing/mm | Welding Speed/mm/s |
|---|---|---|---|---|---|
| #1 | 220, 260, 300, 340 | 1800 | −2 | 4.5 | 9 |
| #2 | 260, 280, 300, 320 | 1800 | −2 | 4.5 | 9 |
| #3 | 220 | 1800 | −2 | 2.5, 3.5, 4.5 | 9 |
| #4 | 220 | 1600, 1700, 1800, 1900 | −2 | 4.5 | 9 |
| #5 | 220 | 1800 | −2, 0, 2 | 4.5 | 9 |
| #6 | 220 | 1800 | −2 | 4.5 | 8, 9, 10, 11 |
| #7 | 320 | 1700 | −2 | 4 | 11 |

## 3. Results and Discussions

Pseudo-color processing in Xiris WeldStudio™® software (2.0.3, Xiris Automation, Burlington, ON, Canada) can convert the required Gray-scale images from the CCD camera to pseudo-color images, which will assist in analyzing the detailed information of arc profile rather than temperature. Figure 3a,b show that the arc shape of laser K-TIG hybrid welding can be divided into three parts: K-TIG arc, laser beam, and coupling arc. Meanwhile, Figure 3c,d depicted macro morphology according to experimental parameter Set #1 in Table 2, when K-TIG welding currents is 220 A.

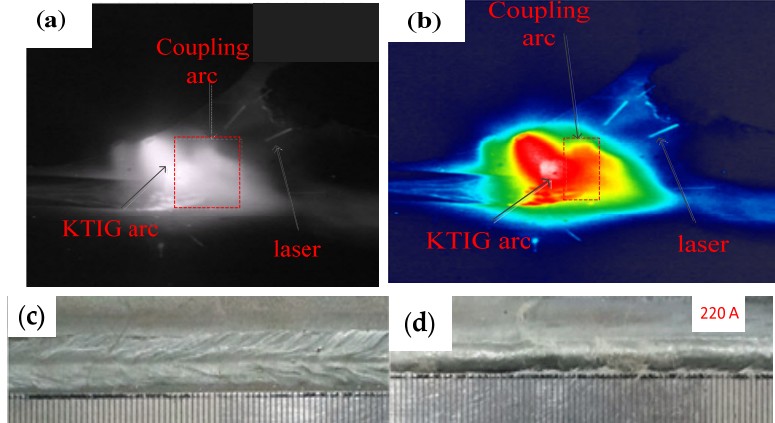

**Figure 3.** Arc profile of laser K-TIG hybrid welding and macro morphology: (**a**) gray-scale image, (**b**) pseudocolor image, (**c**) macro morphology of the front weld, (**d**) macro morphology of the reverse weld.

### 3.1. The Effect of K-TIG Welding Current on the Laser K-TIG Arc Profile

To explore the effect of K-TIG welding current on the laser K-TIG process, the successive images of the hybrid arc profile are shown in Figure 4, according to experimental parameter Set #1 in Table 2. As shown in Figure 4, the arc profile was changed at different degrees with the increase of the K-TIG welding current.

Figure 4 illustrates that K-TIG arcs are all shifted to the direction of the laser beam, and the laser has a certain guiding effect on the K-TIG arc. This is because a high-energy-density laser incident on a metal surface will generate a large amount of metal vapor, and its ionization voltage is much lower than the ionization voltage of gaseous particles, which makes the surface metal easily ionized by heat and forms a photoplasma cloud. The arc will preferentially select the channel with low ionization potential to pass through, so the phenomenon that the K-TIG arc shifts in the direction of the laser beam occurs. However, as the KTIG current increases, the offset decreases. The guiding effect of the laser on the arc is also significantly weakened. This is caused by the gradual increase in the arc blowing

force as the current increases. Meanwhile, the arc becomes more concentrated and straight and spreads less. When the welding current is 340 A, the arc spreading area is the smallest, and the angle between the arc and the horizontal workpiece is completely close to 45°. The analysis shows that when K-TIG welding current is increased enough to form a keyhole, the arc converges at the keyhole, making the arc morphology more concentrated. At this time, if the laser acts on the outside of the keyhole, a serious splash occurs; otherwise, the splash is weakened.

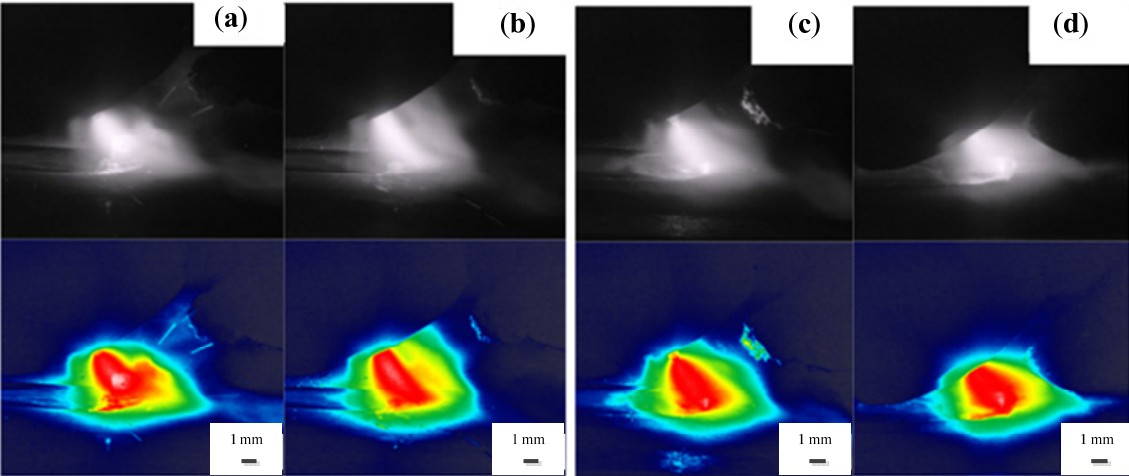

**Figure 4.** Schematic diagram of laser K-TIG hybrid arc profile at different K-TIG welding currents: (**a**) 220 A, (**b**) 260 A, (**c**) 300 A, (**d**) 340 A.

In addition, it is observed that the hybrid arc burns violently after coupling. If laser and arc act together on the keyhole, the penetration depth is further increased.

### 3.2. The Effect of K-TIG Welding Current on the Welded Depth and Width

To explore the effect of K-TIG welding current on the welded depth and width, the successive images of cross-section morphology of welded joints are shown in Figure 5, according to experimental parameter Set #2 in Table 2.

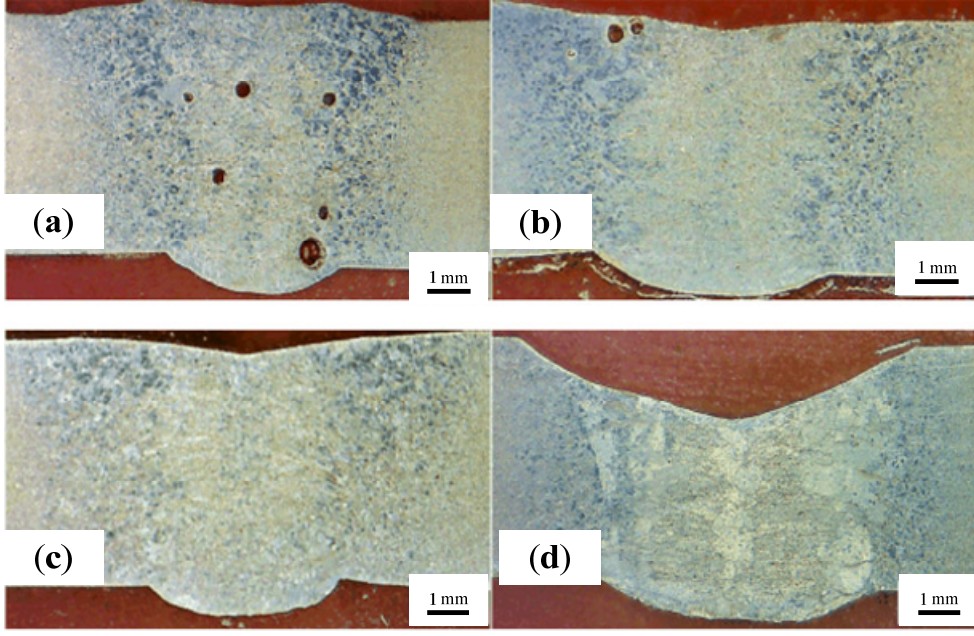

**Figure 5.** The cross-section morphology under different K-TIG welding currents: (**a**) 260 A, (**b**) 280 A, (**c**) 300 A, (**d**) 320 A.

Pore defects appear in Figure 5a. This is because a small welding current of KTIG means poor stability, not enough to resist the interference of the laser on the tig arc, resulting in the generation of pores defects. Figure 5a–c depicted that as K-TIG arc current increases from 260 A to 300 A, the depth and width of welded joints increase slightly, but not obviously. When K-TIG arc current is greater than 300 A, the welded depth and width increase significantly. Meanwhile, it is found that the pore defects are less likely to be generated when arc current is larger than that in the smaller case. That is because the plasma generated by K-TIG arc has a diluting effect on the photoinduced plasma, which improves the absorption rate of the workpiece to laser light. Therefore, with the increase of K-TIG arc current, the depth and width of welded joints increase slowly. On the contrary, when K-TIG arc current reaches 300 A, a keyhole is generated, and the laser acts on the bottom of the keyhole to form a new keyhole, which further increases the weld depth. The arrangement of the arc behind and inclined backward will generate arc blowing force to make liquid metal in the molten pool discharge to the front of the molten pool, which will have a stirring effect on the molten pool and cause the liquid level of the molten pool to sink. Therefore, with the increase of arc current, the stirring effect is enhanced, which promotes the discharge of pores. However, as shown in Figure 5d, when K-TIG welding current is too large, such as 320 A, it also causes the phenomenon of molten pool collapse.

### 3.3. The Effect of the Distance between Heat Sources on the Weld Formation

Figure 6 depicts laser K-TIG arc profile at the different distances between heat sources, according to experiment parameter Set #3 in Table 1. As seen in Figure 6, the laser beam itself has little effect on the arc, but when the distance between the heat sources is less than 3 mm or greater than 5 mm, the laser and the arc become decoupled, resulting in an unstable welding process. This is due to the difference in the distance between the heat sources and the different positions of laser beams.

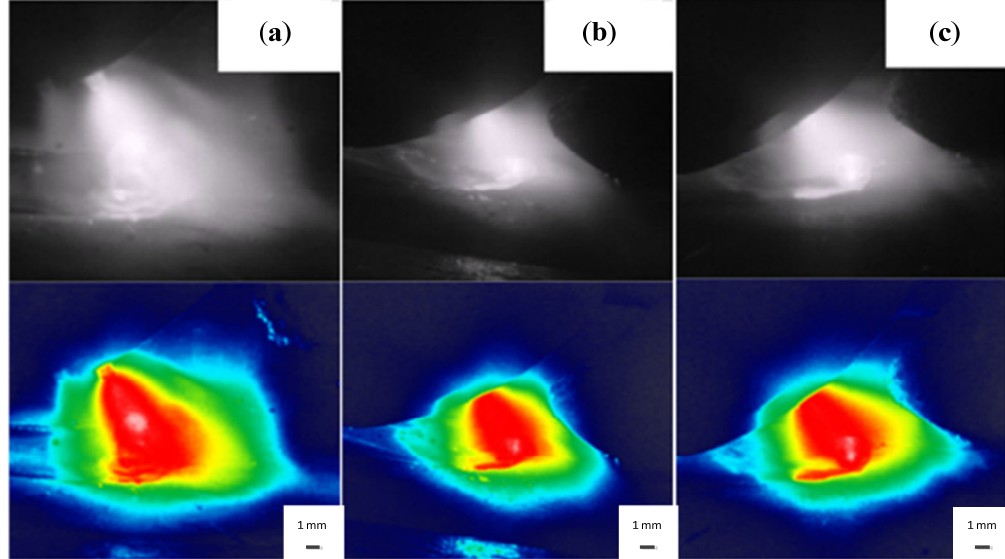

**Figure 6.** Schematic diagram of laser K-TIG hybrid arc profile at different distance between heat sources: (**a**) 2.5 mm, (**b**) 3.5 mm, (**c**) 4.5 mm.

Figure 7 depicts the successive images of cross-sections of welded joints at the different distances between heat sources. When the distance between the heat sources is small (2.5 mm), the cross-sections of welded joints have a small-welded depth and a large-welded width. This is because the laser is very close to the arc, causing the laser to pass through the center of the original channel of the arc. Due to the high plasma concentration in the center of arc, the laser has a certain defocusing effect, thereby reducing the energy density of the laser beam and affecting its penetration. When the distance between heat sources is large

(5.5 mm), the laser and arc become decoupled. The attraction of laser to arc disappears, and the anode spot of the K-TIG arc cannot be stabilized at the place where the laser is applied. The two heat sources are not only unable to cooperate but also interfere with each other to produce splashes. When heat sources' spacing is between 3.5–4.5 mm, the upper part of welded structure is in a TIG welding state (large heat-affected zone), while the lower part is in a laser welded state (small heat-affected zone). However, the overall welded depth has increased. In addition, it was found that when the distance between heat sources was 2.5 mm and 3.5 mm, porosity defects appeared in welded joints.

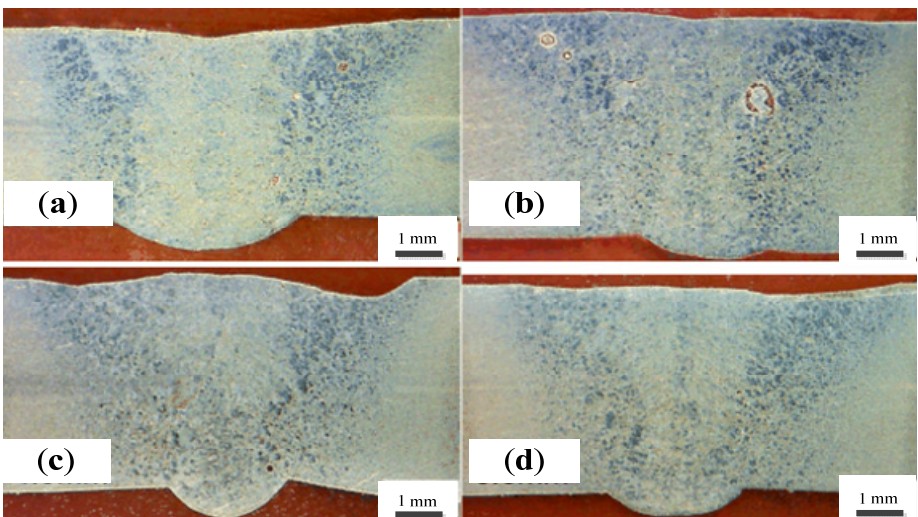

**Figure 7.** The cross-section morphology under different distance between heat sources: (**a**) 2.5 mm, (**b**) 3.5 mm, (**c**) 4.5 mm, (**d**) 5.5 mm.

This is because the different distances between heat sources result in different positions of the laser beam incident on the molten pool. When the distance is small, the laser beam acts on the center of the molten pool. The high plasma density of arc affects the absorption of the laser by the molten pool. In addition, the laser beam may also hit the tungsten electrode and cause energy loss. When the distance is moderate, the laser beam acts on the bottom of the keyhole. The arc plasma has a good dilution effect on the photo-induced plasma. Meanwhile, the laser-arc synergistic coupling effect is the best, thereby further increasing the welded depth. When the distance is larger, the photoplasma and arc plasma begin to separate resulting in poor coupling. In addition, the absorption of laser light by the workpiece is also weakened.

### 3.4. The Effect of the Laser Power on the Welded Depth and Width

To explore the effect of the laser power on the welded depth and width, the successive images of cross-sections of welded joints are shown in Figure 8, according to experimental parameter Set #4 in Table 2.

As seen in Figure 8, with the increase of the laser power, the welded depth gradually increases. Especially when the laser power is greater than 1700 W, the welded depth increases significantly. This is because laser power directly determines the plasma size and intensity. When it is less than 1700 W, the plasma volume is small, and the strength is weak, resulting in an insignificant coupling effect. As the laser power increases, the volume and intensity of the plasma increase. Especially, when the laser power is greater than a certain power (1700 W), a large amount of metal vapor is ionized to form a photoinduced plasma, and then, the photoinduced plasma enters the conductive channel of K-TIG arc, making the arc burn more violently and further promoting the deepening of the K-TIG keyhole. In addition, if the laser power continues to increase, the density of the photoplasma generated by laser radiation on the surface of the workpiece also increases. Meanwhile, the dilution effect of K-TIG arc plasma on the photoplasma is weakened. The scattering effect reduces

the absorption of laser light by the workpiece, resulting in a decrease in the welded depth of hybrid welding. At the same time, it is found that the welded width does not change much, which indicates that the laser power has no obvious effect on welded width.

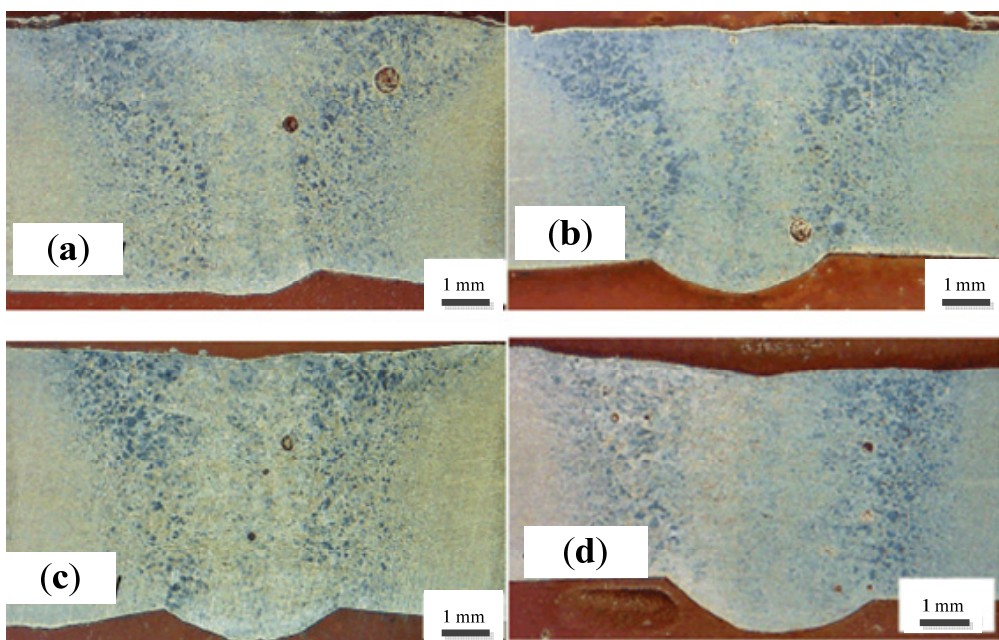

**Figure 8.** The cross-section morphology under different laser power: (**a**) 1600 W, (**b**) 1700 W, (**c**) 1800 W, (**d**) 1900 W.

*3.5. The Effect of the Laser Defocusing and Welding Speed on the Welded Depth and Width*

To explore the effect of the laser defocusing and welding speed on the welded depth and width, the successive images of cross-sections of welded joints are shown in Figures 9 and 10, according to experimental parameter Set #5 and Set #6 in Table 2, respectively.

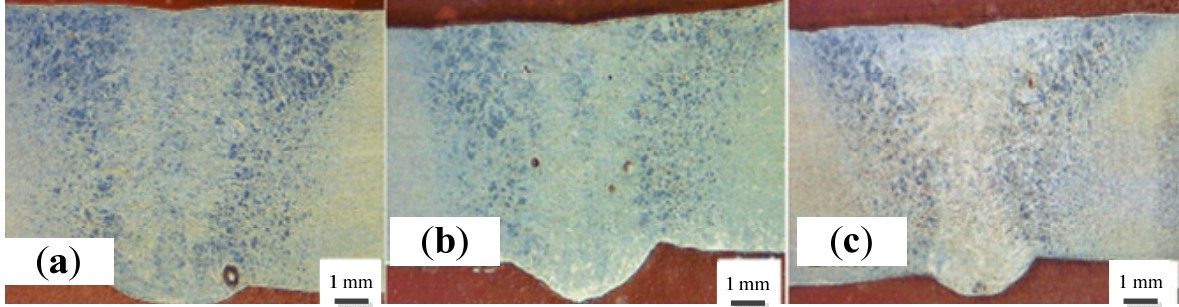

**Figure 9.** The cross-section morphology under different laser defocus: (**a**) −2 mm, (**b**) 0 mm, (**c**) 2 mm.

The defocusing amount refers to the position of the laser focusing relative to the surface of workpiece to be welded. When the laser focusing is above workpiece, it is defined as positive defocus; otherwise, it is defined as negative defocus. As seen in Figure 9, when defocusing amount is −2 mm, the welded depth is the largest. At this moment, laser acts on the bottom of K-TIG keyhole to further increase penetration depth. When the defocus amount is 0 mm, the welded depth is larger. This is because laser acts on the middle of K-TIG keyhole, and the welded depth also increases to a certain extent. When the defocusing amount is 2 mm, the welded depth is obviously reduced, and the cross-sections of welded joints are changed. This is because laser focus is above the molten pool and cannot directly act on the workpiece. The change of the defocus amount will have an effect on the spot size, resulting in the change of the laser power density acting on the surface of

the workpiece, which affects workpiece's absorption of the laser. For best hybrid welding results, the defocus amount should be set at about −2 mm.

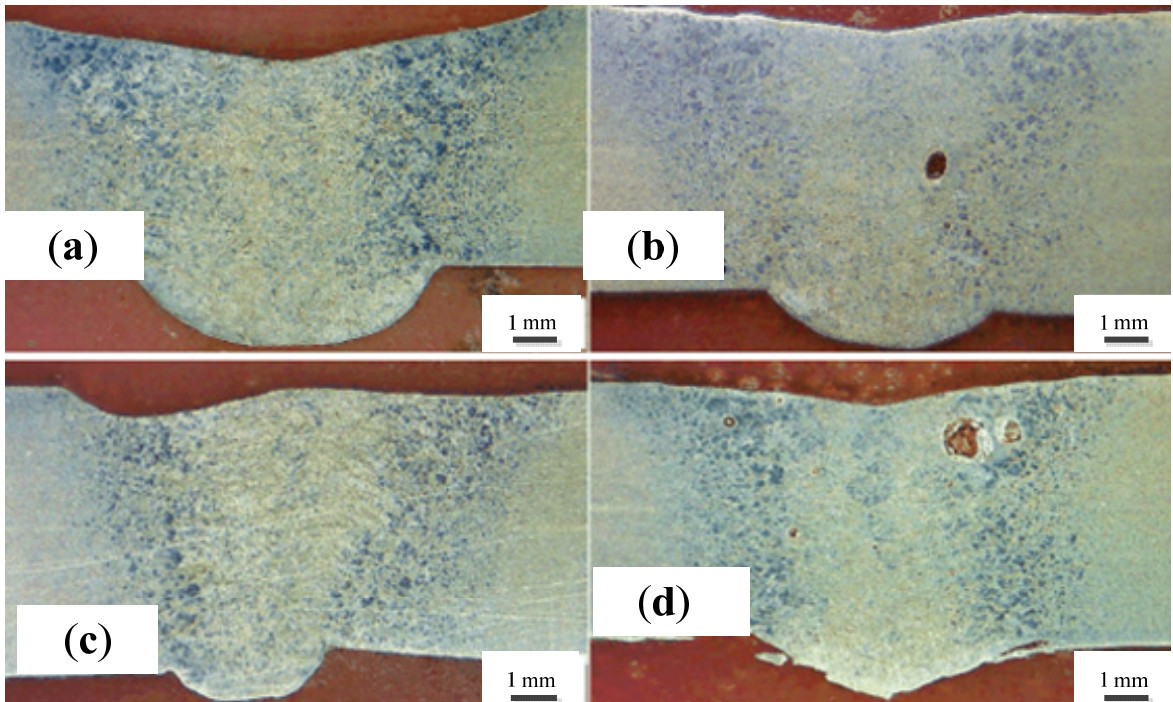

**Figure 10.** The cross-section morphology under different welding speed: (**a**) 8 mm/s, (**b**) 9 mm/s, (**c**) 10 mm/s, (**d**) 11 mm/s.

Figure 10 depicts the cross-section of welded joints at different welding speeds. As shown in Figure 10, the weld depth decreases with increasing welding speed. The reasons are as follows: Firstly, the increase of welding speed will cause the decrease of hybrid welding line energy, which will inevitably reduce the welded depth. Secondly, with the increase of welding speed, the weakening of arc blowing force is also one of the reasons for the decrease of welded depth. Finally, the increase of welding speed also leads to arc shrinkage, which reduces the range of arc heating area, resulting in a smaller welded depth of hybrid welding. Meanwhile, when the welding speed was less than 8 mm/s, the cross-section of welded joints showed the characteristics of single arc welding. This is because the K-TIG arc generates too much plasma, which absorbs the laser energy, resulting in poor coupling between laser and arc. Therefore, the welding speed not only affects the heat input of the hybrid heat source, but also the interaction of the two. Considering the high arc current of K-TIG and its large heat input, to ensure laser K-TIG coupling effect, increase the welded depth and improve the efficiency, the welding speed of about 10 mm/s should be maintained in hybrid welding.

## 4. Variation in Microstructure and Mechanical Properties

### 4.1. Microstructure

Figure 11 depicts the microstructure distribution of the base metal and a typical laser K-TIG weld. As seen in Figure 11, the microstructure of laser K-TIG hybrid welding seam is divided into arc action zone and laser action zone. The K-TIG arc energy is concentrated in the upper part of the molten pool. The welding seam has a large-welded width and a small-welded depth, and the shape of welding seam is funnel-shaped, similar to that of a single TIG welding. The laser runs through the entire weld, so the lower part of the molten pool only has the effect of laser, and the weld width is narrow, showing an inverted cone.

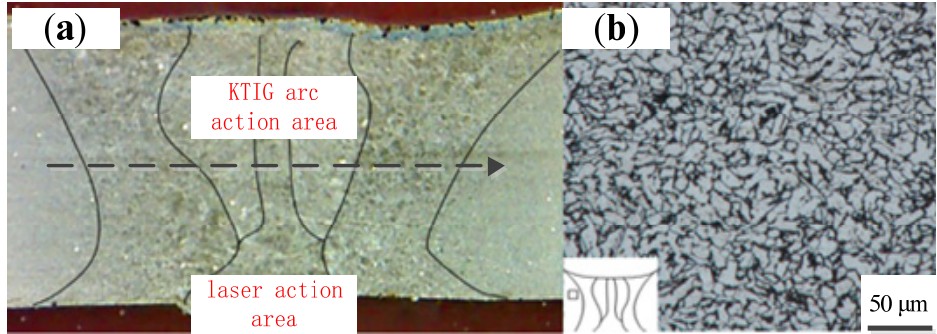

**Figure 11.** The microstructure distribution: (**a**) laser K-TIG hybrid weld microstructure, (**b**) the base metal microstructure.

According to experimental parameter Set #7 in Table 2, the metallographic inspection was carried out. Figure 12 depicts the microstructure distribution of the weld center, fusion zone, and heat-affected zone in K-TIG arc action area. Figure 12a shows that the center of the weld is mainly reticulated equiaxed grains with different sizes. Figure 12b shows that the larger grains in the heat-affected zone of the K-TIG weld have recrystallized and grown, with local thick dendritic precipitates. The fusion zone is partly composed of cellular crystals and elongated columnar crystals, and the columnar crystals mainly grow to the middle and upper part of the weld. Figure 12c is the coarse-grained part, and it can be seen that the grown and formed cellular crystals. Figure 12d is the fine-grained part and not fully grown. The grains in the entire heat-affected zone tend to increase from small to large, and the grains are coarser as they approach the center of the weld. This is because K-TIG current input is large during hybrid welding, resulting in a difference in the size of grains in the heat-affected zone along the center of the weld. At the same time, for the upper weld, the two heat sources work together, so the area of the heat-affected zone is larger, and the trend of grain growth is more obvious. For the lower weld, the laser heat source plays a major role, so the heat-affected zone is small, and the trend of grain growth is relatively slow.

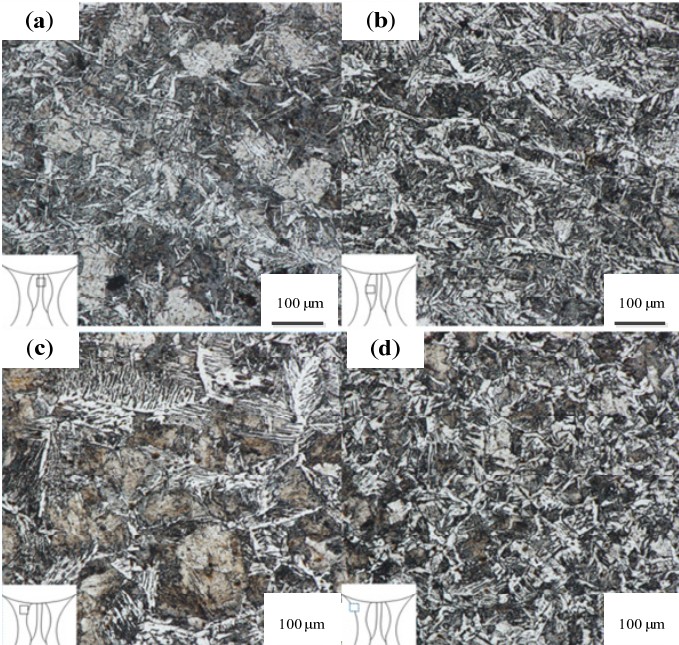

**Figure 12.** Weld microstructure: (**a**) weld center microstructure, (**b**) fusion zone microstructure, (**c**) coarse-grained microstructure of heat-affected zone, (**d**) fine-grained microstructure of heat-affected zone.

*4.2. Hardness Profile and Tensile Test*

Figure 13 describes the microhardness distribution of welded joints, according to experimental parameter Set #7 in Table 2. As shown in Figure 13, the black arrow shows the direction and location of hardness measurements. By comparison, the microhardness of hybrid welding seam increases with the center of weld to the left and right sides. The maximum hardness value is reached near the fusion line, and then, it decreases rapidly and stabilizes. The average microhardness of the base metal is measured to be 155 HV, and the microhardness of hybrid welding seam is greater than that of the base metal, with a maximum of 220 HV.

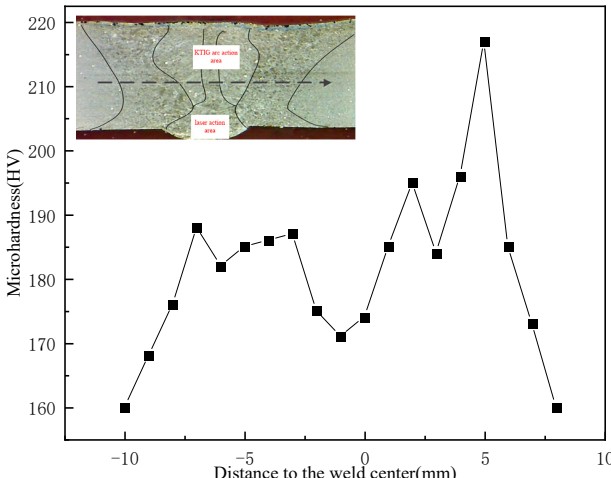

**Figure 13.** Microhardness profile on cross-sections of weld joints.

Tensile specimens were made from hybrid welding seams obtained under different arc currents, heat source distance, welding speed, laser power, and defocusing amount. A total of 44 tensile specimens were obtained. Some samples were selected for the tensile test. The fractures are all located at the base metal position. To measure its actual mechanical properties, a total of 23 tensile specimens were obtained after thinning the middle position of the remaining tensile specimens. Then the tensile test was carried out, and the results were described in Figure 14.

Figure 14a shows that with the increase of K-TIG arc current, the tensile strength first increases and then decreases. When the KIG arc current was 300 A, the tensile strength reached 495 MPa. When the K-TIG current was 320 A, the tensile strength decreased by 7.9%. This is because the arc is behind and tilted backward, which will generate arc blowing force to discharge the liquid metal in the molten pool to the front of the molten pool. When the current increases to a certain level, the arc blowing force increases, and the heat input increases, causing the molten pool to sink, resulting in a sharp decrease in tensile stress. Figure 14b depicts that the tensile strength first increases and then decreases as the heat source spacing increases. When the distance between the heat sources is 3.5 mm, the tensile strength reaches a maximum of 580 MPa. Figure 14c shows that as the welding speed increases, the tensile strength first increases and then decreases. When the welding speed is 11 mm/s, the tensile strength reaches a maximum of 530 MPa. When the welding speed is 12 mm/s, the tensile strength decreases but still reaches 510 MPa. This is mainly due to the uneven heat input of welding. If the welding speed is lower than 10 mm/s, the welding heat input will be too large, which will make the weld grains coarse and the performance decrease. Therefore, laser K-TIG hybrid welding is more suitable for high-speed welding. Figure 14d depicts that the tensile strength generally increases with the increase of laser power. This is because the increase of laser power will generate intense photoplasma, which will enter the arc conduction channel and make the arc burn more intensely. Therefore, the increase of laser power further promotes the deepening of the K-TIG keyhole. Figure 14e shows that the tensile strength decreases sharply as the laser

defocusing becomes larger. When the defocusing amount is −2 mm, the tensile strength is the largest, up to 530 MPa. When the defocusing amount is 0 mm, the tensile strength decreases by 13%. This is because when the defocusing amount is −2 mm, the place where energy density is strongest is inside the base metal. At the same time, laser focusing will be slightly lower than the bottom of the molten pool and laser energy density is the highest in this section. This helps to form laser pinhole welding, achieve the best hybrid welding effect, and enhance the mechanical properties of the weld while increasing welded depth.

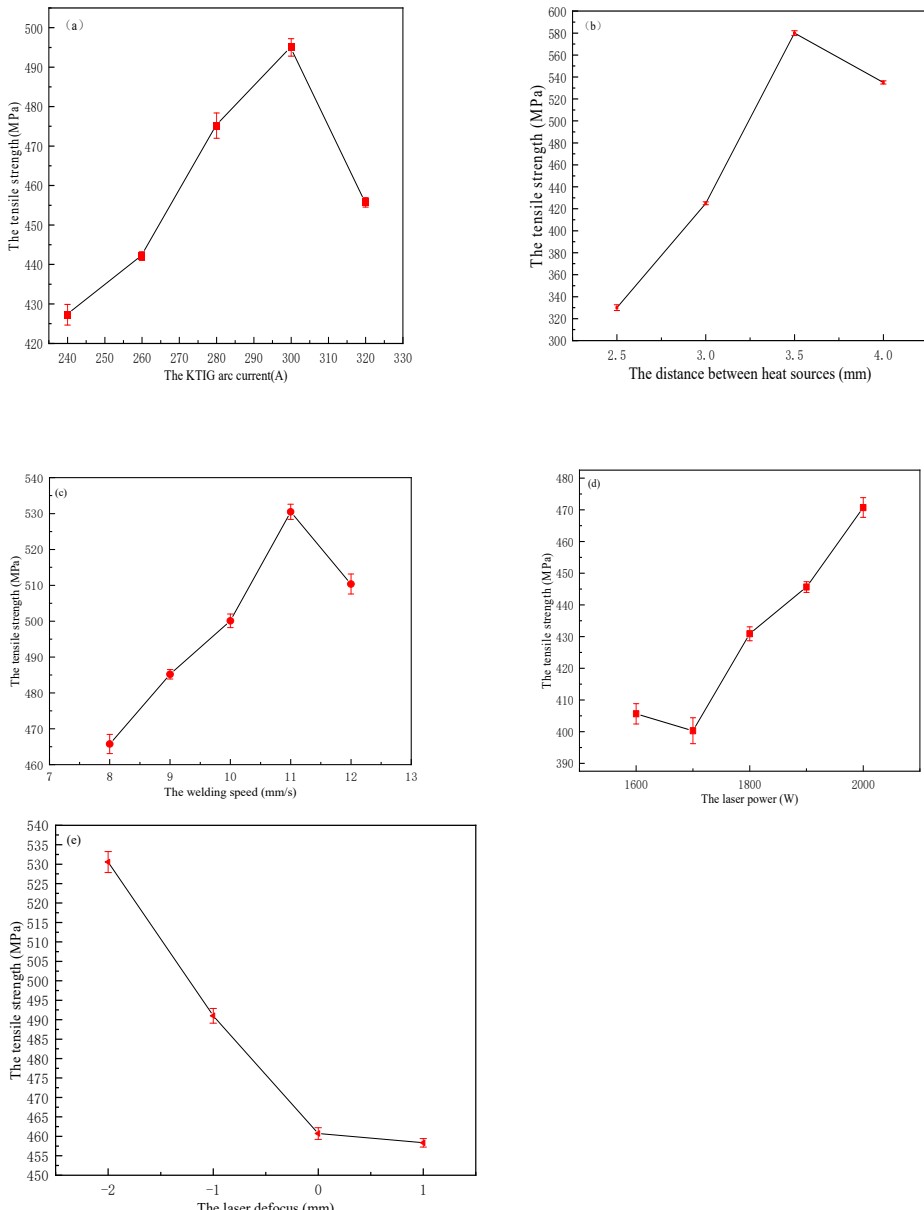

**Figure 14.** The tensile strength at different hybrid welding parameters: (**a**) the K-TIG arc current, (**b**) the distance between heat sources, (**c**) the welding speed, (**d**) the laser power, (**e**) the laser defocus.

## 5. Conclusions

The conclusions were as follows:

(1)　A laser K-TIG hybrid welding test system has been successfully built, and through this system, the parameters of hybrid welding such as laser power $P$, welding current $I$, laser defocusing amount $\delta f$, welding speed $V$, and heat source distance $D$ can be adjusted. Stable, flat-plate-butt experiments were carried out on Q235 carbon steel.

(2) The addition of laser makes K-TIG arc offset, and as K-TIG arc current increases, the offset becomes smaller. If K-TIG arc current and laser power are too large, a large amount of photoinduced plasma is generated to shield laser energy and the welded depth decreases. Different heat source distances and laser defocusing amounts will cause the laser beam to act at different positions. When the laser acts on the bottom of K-TIG keyhole, the welded depth can be further increased. Due to the coupling effect between the heat source and the large K-TIG current, a high welding speed should be maintained during laser K-TIG hybrid welding process to prevent the heat input from being too large and the molten pool from collapsing.

(3) The microstructure analysis and microhardness test of typical welded of laser K-TIG hybrid welding show that the welded morphology is a nail shape with a wide upper part and a narrow lower part. The crystalline direction of the weld seam is perpendicular to the fusion line and points to the center line of weld seam. The grains in the action zone (upper part) are coarser than those in the laser action zone (lower part), and the fusion zone shows, more columnar crystals. In addition, the heat affected zone in the laser action zone is small, and almost no change in the structure near the weld can be seen.

(4) The tensile test found that the fractures were all located at the base metal position. In order to measure the actual mechanical properties of weld, the samples were reworked, and the influence of hybrid welding process parameters on the tensile strength was studied. The weld strength first increases and then decreases with the increase of K-TIG arc current $I$, heat source distance $D$, and welding speed $V$, respectively. The weld strength first decreased and then increased with the increase of laser power $P$, and it showed a downward trend with the increase of defocusing amount $\delta f$.

**Author Contributions:** Conceptualization, J.Y., Z.Z., and J.G.; data curation, H.Z.; formal analysis, J.Y.; funding acquisition, Y.L.; investigation, Z.S. (Zhaofang Su); methodology, J.Y. and Z.Z.; resources, Y.L.; supervision, Z.S. (Zhaorong Sun); validation, J.G.; visualization, H.Z.; writing—original draft, H.Z.; writing—review and editing, Y.L. All authors have read and agreed to the published version of the manuscript.

**Funding:** This research was funded by the National Natural Science Foundation of China (52175305).

**Institutional Review Board Statement:** Not applicable.

**Informed Consent Statement:** Not applicable.

**Data Availability Statement:** If readers are interested in the data, please contact the corresponding author for the complete dataset.

**Acknowledgments:** The work was financially supported by National Natural Science Foundation of China (52175305) and the central government of Shandong province guide local science and technology development fund project (YDZX20203700003578) and 2021 Major industrial key project of transformation of old and new driving forces in Shandong Province.

**Conflicts of Interest:** The authors declare no conflict of interest.

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
