# Peer review of "Arc Characteristics and Welding Process of Laser K-TIG Hybrid Welding"

_metals, doi:10.3390/met12071139_

Round 1
Reviewer 1 Report
Dear Authors,
The article is interesting.
I have to sudjest only couple of technical issues:
1. The schematic diagrams of the laser hybrid arc profiles, for example on Figs. 3, 4 and 6, do not have any scale bar shown. This is not possible to compare the size of the weldings (spots). I advice to add scale bars;
2. The next issue is with same laser arc profiles. There is no designation of the colors mapping. It is not possible to compare the temperatures. Please add the color maps and add description to the text;
3. The measured hardnesses and tensile strengths are shown with precision of two digits after comma, for example Part 4.2 on page 10. It is meaningless to show the hardness and tensile strength testing results with such a precision. Please correct according to the standards.
Best regards,
Reviewer 2 Report
1. What is the purpose of proposing hybrid welding using two high energy welding methods, which can make deep penetrations, justify?
2. Please provide the weld bead appearance, based on the Fig 2, torch angles have much influence on the weld bead formation and penetration control.
3. What is the action of laser welding source on molten pool, it seems to be that the solidifying weld bead again remelts by laser beam?
4. Is the base metal is coated with any kind of primer coats?
5. The presence of pores seems to be due to the KTIG arc wondering by laser arc, please conform it.
6. Please provide the fractography results to identify the modes of fractures.
7. What is maximum thickness can be welded using this hybrid welded method is needs to be given.
8. The scientific discussion of the manuscript should be improved.
Reviewer 3 Report
Please check the following:
Line 83: Please replace text „45 degrees“ with „45°“. Please change this in all text.
Table 3. In Column K TIG current please delete letter A in rows 3,4,5 and 6.
Please define the type of gas (pure argon?) and voltage for TIG welding so comparation from the aspect of the heat input could be possible.
Please explain hardness designation HK while on Figure 13 there is HV microhardness axis. Also, wg+hich force was applied for microhardness measurement was applied: 0.2 or other?
Line 314: Please check the „Mpa“, correct is MPa, check this issue in all text.
Reviewer 4 Report
Dear authors,
I have read your interesting manuscript with attention, however it is valuable I have some suggestion of corrections which could help you to improve your paper.
1. The abbreviation K-TIG should be for readers decoded during first use in text. The essence of the K-TIG method is very good described in https://doi.org/10.1016/j.jmatprotec.2016.07.005
Introduction
2. Page 1 line 39 it is very expected to mention in introduction about different hybrid welding methods like Plasma(Keyhole Mode)-MAG (https://doi.org/10.1016/j.jajp.2022.100111) and (https://doi.org/10.1051/mfreview/2020001) where effect of synergistic enhancement of adventages of such single methods work if they work together in hybrid system. Most important observed effect is related with welding performance as measured by the weight of the molten weld metal.
3. Page 2 line 38 "it compensates the welding defect of TIG welding that the arc is unstable with the increase of welding speed" I suggest to mitigate criticism of unstable TIG arc because it is difficult to indicate more stable arc welding method.
4. There is no comment that the joints in Figures 5a and 5d are defective from the point of view of the welding quality assurance system
5. There is no information about the geometry of specimens to be welded, length and width (of the plates) and it is not known how long the welded joints were tested and where the sample was taken for metallographic tests. Such a description is necessary to ensure the repeatability of the experiment. The information on the initial temperature of the samples before welding should also be completed.
6. Page 5 line 146 "joints ; The" a typo
7. What the standard of the HK unit is not described in the hardness value description.
8. Hardness distribution on the cross-section of the welded joint should to be corrected. Measurements of the hardness of heterogeneous structures (in the micro scale) due to the variety of phases and other elements of the microstructure of ferritic-pearlitic steel require a statistically reliable measurement. The point on the graph should represent the mean value of 4 measurements and should be enriched with the value of the standard deviation (e.g. T-student distribution). Without the described conditions, it is highly probable that the value in the graph is not representative for a given area. Although it is difficult to deny that there was a trend of changes in all three charts.
9. The required tensile test should be more statistically reliable. Measuring the tensile strength on the basis of one sample (at a point) may be random. Additionally, it was observed in Fig. 14 that the cracks were located in the native material (and although it is good from a practical point of view), it does not allow for the comparison of the welds because they are stronger than the parent material.
Regards
